# The Self-Management Experiences of Adolescents with Type 1 Diabetes: A Descriptive Phenomenology Study

**DOI:** 10.3390/ijerph17145132

**Published:** 2020-07-16

**Authors:** Li-Chen Hung, Chu-Yu Huang, Fu-Sung Lo, Su-Fen Cheng

**Affiliations:** 1College of Nursing, National Taipei University of Nursing and Health Sciences, Taipei 11219, Taiwan; ginnahung@mail.cgust.edu.tw; 2Department of Nursing, Chang Gung University of Science and Technology, Taoyuan 33303, Taiwan; 3School of Nursing, Cedarville University, Cedarville, OH 45314, USA; huangc@cedarville.edu; 4Division of Endocrinology, Department of Pediatrics, Chang Gung Memorial Hospital, College of Medicine, Chang Gung University, Taoyuan 33303, Taiwan; lofusu@cgmh.org.tw; 5Department of Allied Health Education and Digital Learning, National Taipei University of Nursing and Health Sciences, Taipei 11219, Taiwan

**Keywords:** adolescents, type 1, self-management, descriptive phenomenology, Taiwan

## Abstract

Many adolescents with type 1 diabetes experience challenges in achieving good glycemic control and have insufficient understanding in executing interventions for glycemic control. This study aimed to understand self-management experiences of adolescents with type 1 diabetes in Taiwan. In this descriptive phenomenological study, we conducted in-depth interviews with 18 adolescents with type 1 diabetes from the pediatric outpatient clinic of a medical center. Data were analyzed using the Colaizzi’s method. Four themes were identified: (1) misconception regarding self-management of blood glucose; (2) conflict between depending on and breaking away from parental assistance for glycemic control; (3) encounter with disruptions in glycemic control regimen due to the presence of schedule changes; and (4) lack of motivation to achieve good glycemic control. The findings indicated that the misconceptions of adolescents with type 1 diabetes about managing glycemic levels resulted from an insufficient understanding of self-management of diabetes. In Taiwan, the heavy emphasis of academic achievement and changes of schedules during breaks tended to disrupt the regimen for glycemic control. Healthcare professionals are encouraged to provide individualized education focusing on the adolescents’ misconceptions regarding self-management of diabetes.

## 1. Introduction

The International Society for Pediatric and Adolescent Diabetes estimated that the global incidence of type 1 diabetes mellitus (T1D) among children below age 15 was 20/100,000 in 2013 [1]. Approximately 500,000 children are diagnosed with T1D worldwide [2]. In Taiwan, the incidence of T1D in children grew from 3.56 to 5.88 per 100,000 from 1999 to 2010 [3]. Although the incidence of T1D is not as high as that in the Western countries, it has been increasing year by year.

The management of T1D includes several daily insulin injections, measurement of blood glucose (BG) levels 4–6 times daily, regulation of carbohydrate intakes, routine physical activities, and prevention of acute and long-term complications [4]. Adolescents with T1D need knowledge and skills of diabetes care to undertake self-management of diabetes [5,6]. King et al.’s [6] qualitative study showed that the complicated knowledge about diabetes management often caused confusion and misunderstanding and became barriers to their self-management of diabetes. Literature has shown that better self-management in adolescents increased the likelihood of achieving good glycemic control [7,8,9]. However, studies also revealed that many adolescents with T1D did not achieve good glycemic control [10,11,12]. It is critical for healthcare providers to gain an in-depth insight into the experience of adolescents with T1D self-management of blood glucose.

Studies have showed that adolescents with T1D demonstrated poor performance in administering insulin injections, monitoring blood glucose, adhering to meal plans, and engaging in appropriate exercises [12,13]. A strong correlation between psychological factors and self-management of diabetes among adolescents with T1D was found [14]. Negative emotional distress, such as anxiety and depression [15,16,17] and low self-esteem [18,19], decreased actions of adolescents with T1D in managing diabetes. The adolescents’ conflicts with parents [10,20], stress from school, and peer perceptions toward diabetes care also impacted effectiveness of self-management of diabetes [21]. Moreover, the self-efficacy of adolescents with T1D is critical for effective self-management of diabetes Studies also discovered that adolescents with T1D with higher self-efficacy had better self-management of diabetes [11,14]. They were more likely to achieve successful self-management of diabetes when they received more support for independence [13,22].

In Taiwan, adolescents typically have extensively long daily schedules to engage in schoolwork, extra tutoring classes, learn skills, and prepare for relevant certification examinations to improve their performance on the high school/or college entrance examinations. Thus, students in middle school or 12th grade often arrive home after 10 pm. Therefore, the long hours of studying often leaves adolescents with T1D no time to learn from the hospital’s diabetes education lessons, which makes glycemic control challenging. Academic pressure and competition often leave adolescents with insufficient time to learn self-management of diabetes [21]. They frequently rely on their parents to provide relevant knowledge and skills about diabetes care. However, adolescents experience more conflicts with their parents and are often uncertain about how to communicate with their parents [21]. They may disagree with their parents and rebel against parental supervision [5]. These factors may negatively impact their glycemic control. In Taiwan, research on self-management of the adolescents with T1D focuses on school experiences and peer interactions. Challenges from balancing between self-management of diabetes and school works, peer pressure, misconceptions about diabetes impacted effectiveness of self-management of diabetes [23,24]. Lee et al. [25] discovered that adolescents with T1D were ineffective in the following areas when engaging in self-management of diabetes: collaboration with parents, diabetes care activities, diabetes problem-solving, and diabetes related communication. They were significantly lower in their collaboration with parents, and communications with parents and healthcare providers regarding diabetes. In sum, most of the Taiwanese adolescents with T1D experience challenges in self-management of diabetes and have an insufficient understanding of how to integrate glycemic management into their daily routines. Therefore, this phenomenology study investigated the experience of adolescents with T1D in self-management of diabetes under challenges in their growth and development, schooling, and relationship with parents.

## 2. Materials and Methods

A descriptive phenomenological approach was used to understand adolescents’ experiences of self-management of T1D. Phenomenology is a philosophy and a research method designed to explore and understand people’s everyday lived experiences [26]. We used “existential bracketing” [27] to ensure unbiased descriptions of the data and approach the topic without any presumptions regarding adolescents’ experiences of self-management of T1D.

### 2.1. Participants and Setting

Purposive sampling methods were used to recruit 18 participants from the pediatric metabolism/endocrinology outpatient clinic of a medical center in Northern Taiwan. The inclusion criteria included: (1) adolescents aged 13–18 diagnosed with T1D for more than one year and have been regularly receiving insulin treatment, and (2) cognitively competent and capable of communication.

### 2.2. Data Collection

Audio-recorded in-depth interviews were conducted using a semi-structured interview guide (Table 1) which was developed based on the literature of adolescents with T1D and discussions with a panel of experts (one pediatric endocrinologist, three experts prepared by doctors in qualitative research, and one of the experts had a family member with T1D). The first author provided explanations about the study’s purpose to adolescents and their parents and secured consents from parents and assents from adolescents. Then, the first author scheduled an appointment for interviews and determined an interview location with participants; the first author conducted all interviews with the participants. The first author is a well-known healthcare provider to the pediatric patients and parents because of her extensive practice experience in caring for children with diabetes. Additionally, she holds membership in the Taiwanese Association of Diabetic Children and has been serving as a staff for the organization for many years. Most participants choose to be interviewed at locations close to home or nearby fast-food restaurants. Sisters of two participants took part in the interview. The rest of the 18 participants were individually interviewed. Prior to interviews, the participants provided their demographic information and HbA1c levels. These face-to-face interviews lasted between 50–90 min. The recruitment process was concluded when no new themes emerged, which indicated data saturation.

### 2.3. Data Analysis

Colaizzi’s [28] method for data analysis was used to understand the adolescents’ experiences of self-management of T1D. Following the philosophy of Husserl [28], Colaizzi argued that researchers should present data clearly to derive phenomenological information. All interviews were conducted by the first author and transcribed verbatim within 48 h of their completion. The authors kept field notes to allow for later reflections. Three of the authors (L.C., C.Y., and S.F.) regularly met and engaged in the following process: (1) carefully read the transcripts to obtain a sense of the whole content; (2) extract significant statements; (3) formulate meaningful sentences and meanings; (4) analyze and synthesize common characteristics to form categories, clusters of themes and to organize similar conceptual themes to establish “themes”; (5) integrate the findings into an exhaustive description of the phenomenon being researched; (6) establish the fundamental structure of the phenomenon; and (7) show the findings to the participants for validation. The authors read and reread the interview transcripts to explore the implicit meanings in order to discover the essence of the perceived the self-management experiences of adolescents with T1D. No data management software was used.

### 2.4. Methodological Rigor

Four criteria were used to evaluate trustworthiness: credibility, transferability, dependability, and confirmability [29]. To establish credibility, the authors used analyst triangulation by continually engaging seasoned nursing experts and peers in the data analysis process to ensure the results truly reflected the participants’ views. To establish transferability, the authors used a purposive sampling strategy to maximize the range of relevant information and diversity of participants. To ascertain dependability, the audio-recorded, in-depth interviews were transcribed verbatim and verified with the participants to ensure accurate representation of their experiences. The authors also kept a detailed record of steps taken and resolutions made during data collection and analysis with supporting rationales. To establish confirmability, an audit trail was used to ensure findings were based on the participants’ perspectives without the authors’ bias. All authors were involved in the process of data analysis. Regular meetings among the authors throughout the research process were conducted to analyze data and resolve disagreement. Similar contents regarding experiences of self-management of diabetes were categorized together and assigned a code. Similar codes formed a theme. Then, the authors described each theme from the participants’ perspectives. The COREQ is a checklist that provides reporting guidance for qualitative studies [30].

### 2.5. Ethical Considerations

This study was approved by the Chang Gung Medical Foundation Institutional Review Board (No: 201601314B0). Before the interviews, all the participants were briefed on the study procedure and purpose and recruited only after their written consent had been obtained. The participants’ background information was presented number to ensure privacy. All data were used only for analysis in this study.

## 3. Results

### 3.1. Characteristics of Participants

Eighteen adolescents aged 13–18 years (9 males, 9 females) were interviewed. Nine participants were in middle school, whereas nine were in high school. The average years with the T1D diagnosis was 6.48 years (range: 2.0–15.1 year). The average HbA1c was 9.46% (range: 5.7–13.8%). Five participants (27%) reported normal HbA1c (≤7.5%). All participants received insulin four times daily through insulin pens. No parents participated in the interview.

### 3.2. Themes

Four themes were identified: (1) misconception regarding self-management of blood glucose; (2) conflict between depending on and breaking away from parental assistance for glycemic control; (3) encountering disruptions in glycemic control regimen due to the fact of schedule changes; and (4) lack of motivation to achieve good glycemic control (Table 2).

#### 3.2.1. Misconception Regarding Self-Management of Blood Glucose (*n* = 18)

Due to the onset of the disease at young ages, parents learned to provide care and execute the regimen for glycemic control. Therefore, adolescents used the prior experiences taught by their parents for glycemic control. This theme encompasses three subthemes: achieving glycemic control with a fixed dose of insulin, believing in replacing regular exercises with increased physical activity, and believing in achieving better glycemic control by eating less starchy food.

##### Achieving Glycemic Control with a Fixed Dose of Insulin (*n* = 16)

The participants determined the insulin dosages required for each meal based on their prior experiences in what or the same routinely injected dosage. They believed that the glycemic level would not increase as long as they injected insulin [dosages did not matter]. A participant stated:
“I adjust the insulin dosage on my own. I usually use similar daily dosages of insulin …. After middle school, I don’t have a regular mealtime, especially during breaks. So, I use my past experience to estimate a time and dosage of insulin. So far, so good. No problem.”(P #16)

##### Believing in Replacing Regular Exercises with an Increase in Physical Activities (*n* = 18)

The participants believed that glycemic control could be achieved as long as physical activities were increased. They perceived the school physical education courses were adequate for glycemic control. Additionally, they replaced regular exercises with the activities of daily living (ADLs). For example, they increased ADLs by assisting parents with chores. A participant stated:
“I don’t need to exercise as long as I work and stay active. I often walk outdoor. On weekends, I sell coffee with my mom. Because I have to lift things and help with chores, I get to stay active and my blood sugar will not be that high. So, it is fine.”(P #1)

##### Believing in Achieving Better Glycemic Control by Eating Less Starchy Food (*n* = 18)

The participants attributed poor glycemic control to being careless or binge eating. They were uncertain about nutrition calculation and meal replacement and often underestimated the carbohydrate contents in food items. They perceived that a stable blood sugar level could be achieved by controlling appetite and eating less starchy foods. While the middle/and high school adolescents understood the relationship between carbohydrate and blood sugar, they experienced a knowledge deficit in recognizing sugary food items and calculating carbohydrate contents. A participant stated:
“I don’t calculate [carbohydrates]. When I see starchy foods [like mochi or rice], I eat less of them. It’s OK to eat more fruits and vegetables.”(P #3) (*n* = 18)

#### 3.2.2. Conflict between Depending on and Breaking Away from Parental Assistance for Glycemic Control (*n* = 17)

Parents blamed the adolescents for high glycemic levels and began to regulate their eating habits. The participants viewed parental assistance as interference and desired to break away from parental meddling in glycemic management. However, they perceived that reminders, care, and assistance with food preparation from families contributed to better glycemic control. There are three subthemes under theme two: being forced to implement interventions for glycemic control by parents; experiencing parental overprotection; and concealing their behaviors from parents.

##### Being Forced to Implement Interventions for Glycemic Control by Parents (*n* = 17)

The participants attributed their success in executing self-care routines and achieving glycemic control to parents’ constant reminders, assistance in food preparations, and demands to accurately monitor glycemic levels and record dietary contents and insulin dosages. A participant stated:
“Every day, my mother would prepare one extra lunchbox of vegetables. I got tired of eating it after a while. I didn’t want to eat it and thought about throwing it away. But, I am afraid my teacher was found out about it and tell my mom. So, I decided to eat it and then I discovered I did not get hungry that easy. So, I thought I would just do what my mom told me. That way, I would not carelessly eat. So, my blood sugar would not be too high.”(P #5)
“My parents want me to record my blood sugar levels, food I eat, and dosages of insulin because they read the recorded logs. Especially my mom. She would constantly ask: ‘What did you eat today? Did you eat a lot?’ It is very annoying. However, this does reinforce me to regularly monitor my glucose level and do the insulin injection. I would do what I am asked to do.”(P #10)

##### Experiencing Parental Overprotection (*n* = 17)

Parents restricted adolescents from going out due to the fact of their concern about overeating and consequent elevation in glycemic levels. In response to the parental protection, the adolescents disregarded parental supervision and avoided glycemic testing. A participant stated:
“When my mom saw a high blood sugar level, she nagged ‘I told you not to eat carelessly’. She also restricts me from eating. I can’t eat things that I want to eat. Why doesn’t she just lock me up at home so that I would not go out and eat? They thought if I don’t go out, my blood sugar will be better. I am very unhappy about this. So, I don’t want to test my blood sugar anymore.”(P #9)

##### Concealing Their Behaviors from Parents (*n* = 15)

The participants intentionally concealed their behaviors from parents because of their concerns about poor glycemic control and the resulting blames and warnings from parents and healthcare providers. A participant stated:
“I smoke, chew arecas, and drink energy drinks (include medicinal wine). These cause hyperglycemia. I wonder: ‘Doesn’t Pao-Li-Da B (an energy drink in Taiwan) cause hypoglycemia, since it’s wine?’ However, I can’t ask doctors this question. If my dad knows, he would know I drink, and I would be in trouble.”(P #9)
“After I was able to have good control of my blood sugar, I rarely talked to my mom about my blood sugar. I didn’t want my mom to know about my blood sugar level…because my mom would scold me if I didn’t do a good job with controlling my blood sugar. Mom would be unhappy and say: ‘I will not be bothered with you’.”(P #1)

#### 3.2.3. Encounter Disruption in Glycemic Control Regimen Due to the Fac of Schedule Changes (*n* = 16)

The busy schedule in the middle/high schools made it unfeasible for the adolescents with diabetes to test glycemic levels and calculate insulin dosages on time. Additionally, their daily schedules were different during the winter and summer breaks which impacted on the blood sugar monitoring routine and attributed to unstable glycemic levels. There are two subthemes under theme three: being forced to modify the executing routine due to the presence of academic stress and executing routine interventions with decreased frequency due to the changes in the daily schedule during the winter and summer breaks.

##### Being Forced to Modify the Executing Routine Due to the Presence of Academic Stress (*n* = 16)

The middle/high school adolescents had more school examinations and assignments. In addition to regular school time, many adolescents attended cram school to prepare for the high school/college entrance examinations which significantly increased the frequency of eating out. Moreover, they consumed more food to relieve stress from school, which impaired the dietary routine for glycemic control. A participant stated:
“In high school, I have lots of assignments, quizzes, and reports. I often become nervous about them. Also, I have been involved in extracurricular activities and eaten more than I should have. I just cannot control myself.”(P #14)
“Being in high school, I need to get ready for the college entrance exam. I need to go to cram school and study every day. Often, I feel very tired and hungry. So, I just bought food and eat it without thinking about the portions or testing my blood sugar. Sometimes, I forgot to have my insulin before bedtime. I just went to bed. My blood sugar has been high, but there is nothing I can do about it.”(P #17)

##### Executing Routine Interventions with Decreased Frequency Due to the Changes in Daily Schedule during the Winter and Summer Breaks (*n* = 15)

During the winter and summer breaks, the participants were preoccupied by internet surfing, video games, social media or binge-watching TV shows until midnight and arose at noon the next day. Therefore, they injected insulin 2–3 times daily or omitted the long-acting insulin before bedtime. Changes in schedules changed the treatment routine and contributed to poor glycemic control. A participant indicated:
“I did not regularly inject insulin since the summer break in middle school. I constantly played with my cell phone and was unable to get up in the morning to eat breakfast. I would sleep until afternoon. So, I was not able to inject insulin at a regular time. Sometimes, two to three hours after eating a meal, I would eat the next meal. Then, I would eat dinner at midnight. So, I rarely injected the long-acting insulin.”(P #9)

#### 3.2.4. Lack Motivation to Achieve Good Glycemic Control (*n* = 16)

The adolescents were tired of the tedious daily glycemic regimen. They overlooked damages from hyperglycemia since they had never experienced its life-threatening impact. Sometimes, they felt incapable of achieving the expected glycemic level, became passive due to the fear of blame, and thus lacked motivation toward achieving successful glycemic control. There are three subthemes under this theme: being unaware of the seriousness of hyperglycemia; feeling tired of glycemic control regime; and seeking a sense of belonging from peers and avoiding execution of glycemic control regimen.

##### Being Unaware of the Seriousness of Hyperglycemia (*n* = 15)

With hypoglycemia, the adolescents experienced physical discomfort and sensed an immediate threat to life. However, hyperglycemia does not cause immediate discomfort. Only one adolescent experienced discomfort from ketoacidosis. Therefore, most participants only focused on hypoglycemia and ignored the damages of hyperglycemia. A Participants indicated:
“I am uncomfortable when my HbA1c is above 10. I feel fine as long as I don’t have ketoacidosis. So, I don’t usually keep eyes on blood sugar. I eat whenever I want to eat.”(P #9)

##### Feeling Tired of Glycemic Control Regimen (*n* = 16)

The participants carried medical equipment (e.g., a blood glucose meter, a notebook for recording, an insulin pen, and needles) wherever they went, and had to calculate the carbohydrate amount and portions for meal replacement. They grew weary about these tedious tasks, became unmotivated to execute the regimen, and often forgot to inject insulin which contributed to poor glycemic control. A participant stated:
“Since middle school, my mother wanted me to test my blood glucose and inject insulin. But I have been lazy to test blood sugar and often forgotten about insulin injection. I have not regularly injected insulin since 8th grade. When I eat out, I don’t calculate portions. The calculation is cumbersome. I often forgot to do it and went ahead to eat the meal.”(P #7)

##### Seeking a Sense of Belonging from Peers and Avoiding Execution of Glycemic Control Regimen (*n* = 15)

The adolescents desired to assimilate with their peers in school. Therefore, they worried about peer rejection because of the T1D diagnosis. Some adolescents concealed the diagnosis of diabetes or omitted the glycemic control regimen so that they might socialize or participate in student organization activities with their peers, which contributed to poor glycemic control. A participant said:
“In high school, my classmates did not know that I had diabetes. If I told my classmate that I had diabetes, they would be concerned when interacting with me. So, they would not dare to eat in front of me even if they wanted to eat. They would hide from me. I feel isolated. This makes me feel bad. I don’t like to be different from my friends.”(P #6)
“I began to smoke with my friends in middle school. I often went out with my friends to have fun. When I was out with my friends, I did not inject insulin...I am part of a temple worship group. If the group members knew I have diabetes, I would be restricted from participating in the group performance. So, I would not inject insulin when I had performances.”(P #9)

## 4. Discussion

### 4.1. Misconceptions Regarding Self-Management of Blood Glucose

This study showed that adolescents with T1D received diabetes education from their parents, which is similar to findings from Babler et al. [5] study. Both studies reported that adolescents learned diabetes care from their parents. Babler et al. [5] interviewed 15 adolescents to understand their experiences of successful self-management of T1D. In addition to learning from parents’ approaches for diabetes care, they also needed to develop strategies for self-management of diabetes in order to achieve successful glycemic control. Additionally, in this study, the adolescents with T1D believed that their glycemic level would be under control if they received insulin regularly, moved more, and ate less. However, the adolescents with T1D received fragmented knowledge about diabetes. As the adolescents with T1D ate out with peers more often, they often replaced regular meals with the Taiwanese snacks and street food which often contained high amounts of fat and sugar. Additionally, the convenience of fast food restaurants, beverage shops, and stores often hindered their selection of appropriate food. When the adolescents encountered unfamiliar food items, they experienced difficulties in calculating the carbohydrate contents and the needed dosages for insulin. Challenges to achieve good glycemic control among the adolescents with T1D is to be expected. Chilton et al. [21] indicated that successful self-management of diabetes in adolescents with T1D requires their continual engagement in experiential-based learning and abilities to integrate the self-management strategies into their daily lives. Therefore, in addition to calculation of regular meal items, healthcare professionals also need to educate the adolescents with T1D on how to calculate carbohydrate contents of the Taiwanese street food and the required dosages of insulin. This would help the adolescents with T1D to effectively integrate diabetes care into their daily lives. Adolescent’s “food culture” should be taken into consideration when providing diabetes education. Healthcare professionals need to teach them to calculate the carbohydrate amount of their favorite food. Additionally, healthcare professionals may provide multimedia-assisted health education through Internet or application (APP) software to guide adolescent’s learning of glycemic control.

### 4.2. Conflict between Depending on and Breaking Way from Parental Assistance for Glycemic Control

The study involved adolescents with and without good glycemic control. The adolescents achieved better glycemic control with ongoing parental engagement and supervision in diabetes care. In this study, the adolescents were more successful in glycemic control with parental assistance in food preparation. Parents reacted to adolescents’ poor glycemic control with a blaming, judging and nagging approach. The adolescents expressed a sense of frustration towards parents’ forceful approach. Developing independence is a key focus during the adolescence phase. Adolescents fight with their parents for more autonomy and detest parental interferences with their independence. As a result, adolescents may avoid taking responsibilities of self-management of diabetes and lead to unsuccessful glycemic control. Commissariat et al.’s [10] study confirmed that increased conflicts between parents and adolescents may lead to poorer adherence and lower self-efficacy.

While the participants desired to get rid of the parental monitoring and avoid testing blood sugar, they were not seeking complete independence. Rather, they struggled between desiring parental care and seeking autonomy. A participant stated that she burst into tears when her parents told her “I will not be bothered with you” because of her increased glycemic level. Martinez et al. [14] proposed that parents should take adolescents’ capabilities and desires for independence into consideration in order to achieve successful self-management of diabetes. Therefore, parents should continuously be involved in an adolescents’ self-management of diabetes by providing caring reminders with respect to the adolescents’ independence. 

This study highlights Taiwanese parents’ approaches to care for the adolescents with T1D. However, the parental approaches were not what the participants desired. This study does not address the balance between the care needs from the perspectives of adolescents with T1D and the desired approaches for glycemic control from the parental perspectives. Future studies need to also interview the parents in order to gain a comprehensive understanding of the adolescents’ self-management of diabetes.

### 4.3. Encountering Disruption in Glycemic Control Regimen Due to the Fact of Schedule Changes

Middle school and high school schedules are busier and at a higher pace as compared to the elementary school schedule. Recess time decreases from 15 to 10 min. During the 30 min of lunch break, they needed to finish their lunch and clean the school environment. School-time glycemic testing took time away from participating in peer activities. Therefore, they often choose to omit blood sugar testing and simplify the calculation for insulin dosages. This finding is consistent with the findings from Chao et al. [31] and Wang et al. [23] which indicated that adolescents with T1D were unable to balance between school schedules and diabetes care routine. Results from Chao et al.’s [31] study showed that busy life (such as school work, sports, extracurricular activities) created challenges in self-management of diabetes. Additionally, Wang et al. [23] discovered that adolescents may eliminate school-time insulin injection to avoid being different from peers. They often prioritized school schedules over self-management of diabetes which caused poor glycemic control. Therefore, changes in the school schedule may impact on adolescents’ glycemic control.

The Taiwanese educational system places an intense emphasis on academic achievement. Particularly during the middle/high school years, adolescents have significantly more examinations. They become disheartened and stressed if they do not perform at the expected level. Many adolescents choose to alleviate their stress by eating, which impacts their glycemic level. This study also found that adolescents with T1D had more freedom with time during the winter and summer breaks. They spent more time on cell phones, electronic games, social media, and TV dramas and stayed up late. The timing and frequency for insulin injections became disorganized and contributed to poor glycemic control during the winter and summer breaks.

### 4.4. Lack Motivation to Achieve Good Glycemic Control

In the literature, adolescents with T1D who had higher motivation toward independence tended to demonstrate effective diabetes self-management [32,33]. In this study, the adolescents often ignored the severity of the complications from prolonged hyperglycemia and lacked the motivation to actively control glycemic level. The tedious diabetes treatment regimens and adolescents’ desires for peer acceptance often contributed to their lack in motivation for glycemic control. Similarly, Chao et al. [31] discovered that adolescents experienced stress and social embarrassment from the frequency, inconvenience, complexity, and visibility of self-management of diabetes. In this study, the participants avoided to be different from others by hiding from their peers when doing blood sugar testing and insulin injections. 

Reitblat et al. [34] indicated the relationship between healthcare professionals and the adolescents with T1D is critical in promoting motivations towards self-management of diabetes. The more the adolescents sensed the support and respect from the healthcare professionals, the higher the motivation to achieve good glycemic control. This study also found that the adolescents with T1D were motivated to engage in self-management of diabetes when the healthcare professionals acknowledged improvement of HbA1C, praised their efforts, and celebrated success with them. In this study, the adolescents were eager to share their strategies of self-management of diabetes if the researchers began interviews with acknowledging improvement of their HbA1c levels.

### 4.5. Limitations

The setting for this study was a medical center in the Northern Taiwan. Food preparations and preferences may differ between residents in the northern and southern regions. Residents in the northern region are more accustomed to “indoor” shopping due to the increased access to department stores and wholesale grocery stores. Therefore, the findings of this study may be more relevant to the adolescents with T1D living in urban areas. Additionally, only adolescents were interviewed in this study. Parental approaches to assist with self-management of diabetes were reported by the adolescents. More research needs to be conducted to understand the challenges parents experienced in caring for the adolescents with T1D.

## 5. Conclusions

This study found the adolescents’ self-management of diabetes was implemented based on the knowledge and guidance from their parents. The variety of Taiwanese stress food contributed to the adolescents’ inability to effectively calculate the carbohydrate contents and insulin dosages. Additionally, parents’ overprotective approach often contributed to the adolescents’ struggles between independence and dependence. Healthcare professionals may encourage parental participation in adolescents’ self-management of diabetes with consideration of their needs for independence to achieve a mutual goal. Furthermore, adolescents often faced heavy academic loads, pressure from taking the high school/college entrance examinations and peer identity, which may impact effectiveness of self-management of diabetes. Additionally, school-time glycemic testing took time away from participating in peer activities. Furthermore, school nurses need to facilitate teachers to understand the individual needs of adolescents with T1D and provide additional “healthcare time” to allow for sufficient time for glycemic testing and insulin injections. It is also beneficial to form groups that share similar experiences through social media to gain peer support. In addition, only adolescents with T1D were interviewed in this study. The adolescents reported the approaches that their parents utilized to help them with diabetes management. Future research is recommended to investigate the challenges parents experienced in caring for adolescents with T1D from a parent’s perspective. Additionally, studies need to focus on exploring the assistance adolescents desire to receive from parents in order to effectively guide adolescents’ independent self-management of diabetes.

## Figures and Tables

**Table 1 ijerph-17-05132-t001:** Interview guide.

Could you talk about how you to take control of your blood sugar?
What strategies do you usually use to manage your blood sugar? How do you implement these strategies?
Could you share the most effective strategy for blood sugar control? Why is the strategy effective?
Which strategies for blood sugar control are difficult to implement? Why are the strategies difficult to implement?
What problems or difficulties have you encountered when you manage your blood sugar? How have you dealt with these problems or difficulties? Were your problem-solving strategies effective?

**Table 2 ijerph-17-05132-t002:** Summary of themes and sub-themes.

Themes	Sub-Themes
1. Misconception regarding self-management of blood glucose (*n* = 18)	1–1. Achieving glycemic control with a fixed dose of insulin. (*n* = 16)
	1–2. Believing in replacing regular exercises with an increase physical activity. (*n* = 18)
	1–3. Believing in achieving better glycemic control by eating less starchy food. (*n* = 18)
2. Conflict between depending on and breaking away from parental assistance for glycemic control (*n* = 17)	2–1. Being forced to implement interventions for glycemic control by parents. (*n* = 17)
	2–2. Experiencing parental overprotection. (*n* = 17)
	2–3. Concealing their behaviors from parents. (*n* = 15)
3. Encountering disruption in glycemic control regimen due to the fact of schedule changes (*n* = 16)	3–1. Being forced to modify the executing routine due to the fact of academic stress. (*n* = 16)
	3–2. Executing routine interventions with decreased frequency due to the changes in daily schedule during winter and summer breaks. (*n* = 15)
4. Lack of motivation to achieve good glycemic control (*n* = 16)	4–1. Being unaware of the seriousness of hyperglycemia. (*n* = 15)
	4–2. Feeling tired of glycemic control regimen. (*n* = 16)
	4–3. Seeking a sense of belonging from peers and avoiding execution of glycemic control regimen. (*n* = 15)

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
