# Peer review of "The Self-Management Experiences of Adolescents with Type 1 Diabetes: A Descriptive Phenomenology Study"

_ijerph, 2020, doi:10.3390/ijerph17145132_

Round 1

Reviewer 1 Report

Would recommend the use of COREQ checklist to guide the reporting of the study.

Abstract

  • First identified theme (line 21) is written awkwardly and unclear
  • Line 24, final theme: lack OF motivation
  • Lines 24-26: interpretation of findings don't seem to reflect the themes that have been identified 
  • Perhaps more emphasis on misconceptions of diabetes management could be included in the background to provide more context and ensure better link between the research findings and interpretation

Introduction

  • Avoid use of uncommon abbreviations such as AT1D and SMD.
  • Lines 36-37, please provide a reference to support state re increase of T1D incidence per year
  • Line 38, use of "treatment" language. T1D is managed rather than treated. The differentiation between this language use is important as T1D is complex to manage and there are many variables that impact blood sugars. Saying that it's treated diminishes the complexity of its management.
  • Line 41, use of "diabetic care" language. Focus on person centred language, use terms like diabetes care or person with diabetes instead of diabetic.
  • Line 42, check referencing.
  • Line 52, use of "diabetic" language.
  • Lines 59-61 "Therefore, the long hours of studying often leaves AT1D no time to learn from the hospital’s diabetic education lessons, which makes glycemic control challenging." Please provide a reference for this statement, otherwise it reads like an assumption which may not be correct.
  • Lines 61-62 "They frequently rely on their parents to provide relevant knowledge and skills about diabetic care." see above re making assumptions. Also note language.
  • Line 71: most OF the Taiwanese...
  • Line 74: remove "is". 

Methods

  • Disappointing to see that the "expert panel" did not include people with diabetes
  • Lines 91-92 please check grammar and sentence structure
  • Please provide details on who conducted the interviews and their background
  • Table 1: please check formatting
  • Line 100: please expand on Colaizzi's method
  • Strong methodological rigor section - well done

Results

  • Providing the number of participants who expressed a similar theme will be helpful to know how common it was mentioned
  • Results need to be clear that they are derived from the adolescents rather than the parents
  • Line 137: extra parentheses
  • Line 152: unclear that this is a sub-heading
  • Line 158: Too much detail provided about the participant, potentially identifying information 
  • Line 178: "Parents blamed the adolescents for high glycemic levels and began to regulate their eating habits." Is this an assumption or were parents interviewed as well?
  • Line 264: hypoglycaemia and hyperglycemia rather than hypo/hyperglycemic 

Discussion

  • Line 95 - expand on the study from Babler et al. - what did they do and what did they find that was similar
  • Based on the first reported theme, I would also encourage the authors to explore the need for regular diabetes education and the dynamics of these education session in the adolescent years
  • In the second reported theme, casual associations are made that have not been previously presented. Please either include these in the results or omit.
  • Lines 316-320, please support these statements with evidence
  • Lines 336-337 "They might be unable to keep up with their peers if tested glycemic levels at school." please check sentence structure 
  • Line 339 please expand on these studies

Conclusion

  • Lines 383-384 "They might be unable to keep up with their peers if tested glycemic levels at school." expand more on this statement and perhaps focus the conclusion more on future research directions or practice implications as a result from this study.

Author Response

Dear reviewer : We have followed the reviewer's comments and response point by point. Please see the attachment. Thank you so much for your considerations.

Reviewer 2 Report

Reviewer’s report

Title: The Self-Management Experiences of Adolescents with Type 1 Diabetes: A Descriptive Phenomenology Study.

Authors: Li-Chen Hung, Chu‐Yu Huang, Fu-Sung Lo and Su‐Fen Cheng.

General remark:

Adolescence is the most challenging period in the management of type 1 diabetes in young patients. Therefore, studies assessing critical aspects of the functioning of adolescents with type 1 diabetes, are vital to establishing appropriate standards of patients management, including the education of both children and parents. However, before the manuscript is published, some important issues should be clarified.

Major revisions:

Materials and Methods

Was the power calculation performed to establish the number of study participants?

Results

This part of the paper is strictly descriptive. Even though the authors declare the descriptive character of their work in the paper’s title, since it is a scientific paper, they could support their findings by some statistical calculations.

Minor revisions:

Introduction

Lines 33-34 “The International Society for Pediatric and Adolescent Diabetes estimated that the global incidence of type 1 diabetes mellitus (T1D) among children below age 15 was 20/100,000”  – please specify the period the statistics refer to.

Lines 40-41 “Adolescents with T1D (AT1D) need knowledge and skills of diabetic care to undertake self-management of diabetes (SMD) [1,5]; King, King, Nayar, & Wilkes, [5].” –  please remove the authors' names and the doubled number of citations.

Line 47 “Studies have showed” – shouldn't be changed to “Studies have shown”?

Lines 70-71 “In sum, most the Taiwanese AT1D experience challenges in SMD and have insufficient understanding how integrate glycemic management into their daily routines”  – aren't some prepositions missing? ("In sum, most the Taiwanese AT1D experience challenges in SMD and have an insufficient understanding of how to integrate glycemic management into their daily routine")?

Materials and Methods

Table 1. Interview guide

  • Line numbering is unclear, why the first question is treated as the table heading?
  • The question tag “How so?” in questions 4 & 5 is unclear. Please change into the proper one.

Results

Lines 143-144 “Lack motivation to achieve good glycemic control (Table 2).  – isn’t the preposition missing (“Lack of motivation…..”)?

To facilitate the reception, I suggest highlighting the subsection headers, e.g.:

Line 152 “Achieving glycemic control with a fixed dose of insulin.“

Line 159 “Believing in replacing regular exercises with increase physical activities.”

Line 167 “Believing in achieving better glycemic control by eating less starchy food”... end so on.

Author Response

Dear reviewer: We have followed the reviewer's comments and response point by point. Please see the attachment. Thank you so much for your considerations.

Round 2

Reviewer 1 Report

Thank you for your work in addressing the comments. Please ensure that a final proof read is conducted to check for grammatical and formatting errors. Please also ensure that patient-centred language is used. i.e. using person with diabetes rather than diabetic or diabetes care rather than diabetic care.

Author Response

(The authors gave the same response as above.)

Reviewer 2 Report

I want to express my gratitude for the opportunity to re-review the manuscript entitled ”The Self-Management Experiences of Adolescents with Type 1 Diabetes: A Descriptive Phenomenology Study”. The Authors took into account my previous comments regarding the methodology; therefore, in my opinion, the manuscript meets the criteria necessary for publication in the International Journal of Environmental Research and Public Health.

Author Response

Thank you for your recommendations for our manuscript. We greatly appreciate the opportunity to be considered for publication. Best regards.